# Identification, proteolytic activity quantification and biofilm-forming characterization of Gram-positive, proteolytic, psychrotrophic bacteria isolated from cold raw milk

Mehdi Zarei[1]*, Sahar Elmi Anvari[1], Siavash Maktabi[1], Per Erik Joakim Saris[2], Amin Yousefvand[2]*

1 Department of Food Hygiene, Faculty of Veterinary Medicine, Shahid Chamran University of Ahvaz, Ahvaz, Iran, 2 Department of Microbiology, Faculty of Agriculture and Forestry, University of Helsinki, Helsinki, Finland

* zarei@scu.ac.ir (MZ); amin.yousefvand@helsinki.fi (AY)

**Data Availability Statement:** All relevant data are within the manuscript and its Supporting Information files.

## Abstract

Psychrotrophic bacteria of raw milk face the dairy industry with significant spoilage and technological problems due to their ability to produce heat-resistant enzymes and biofilms. Despite extensive information about Gram-negative psychrotrophic bacteria in milk, little is known about Gram-positive psychrotrophic bacteria in milk, and their proteolytic activity and biofilm-forming characteristics. In the present study, Gram-positive, proteolytic, psychrotrophic bacteria of cold raw milk were identified, and their proteolytic activity and biofilm-forming capacity were quantified. In total, 12 genera and 22 species were represented among the bacterial isolates, however 50% belonged to three genera, namely *Staphylococcus* (19.4%), *Bacillus* (16.7%), and *Enterococcus* (13.9%). Different levels of proteolytic activity were detected in the identified isolates, even among the strains belonging to the same species. In addition, proteolytic activity was significantly higher at 25˚C than at 7˚C for all isolates. The crystal violet staining assay in polystyrene microtitre plates revealed a high level of variation in the biofilm-forming capacity at 7˚C. After 72 hours of incubation, 11.1% of the strains did not produce a biofilm, while 27.8%, 52.8%, and 8.3% produced low, moderate, and high amounts of biofilm on polystyrene, respectively. The psychrotrophic bacteria were also able to produce biofilms on the surface of stainless steel coupons in ultra-high temperature milk after 72 h of incubation at 7˚C; the number of attached cells ranged from 1.34 to 5.11 log cfu/cm$^2$. These results expand the knowledge related to the proteolytic activity and biofilm-forming capacity of Gram-positive psychrotrophic milk bacteria.

**Funding:** A research grant (SCU.VF99.245) provided by Shahid Chamran University of Ahvaz supported this study. In addition, The Finnish Food Research Foundation and The Finnish Society of Sciences and Letters are acknowledged for awarding grants to Amin Yousefvand on the dates 16.03.2021 and 28.03.2022. The open access funded by Helsinki University Library. The funders had no role in study design, data collection and analysis, decision to publish, or preparation of the manuscript.

**Competing interests:** The authors declare that there is no conflict of interest regarding the publication of this article.

## Introduction

Cooling and refrigeration have led to significant improvements in the bacteriological quality of raw cow's milk that is stored and transported in bulk. However, a long-term refrigerated storage of raw milk induces psychrotrophic bacteria to dominate the microbiota [1–4]. Although psychotrophic bacteria represent less than 10% of the original raw milk microbiota, they typically account for over 90% of the total microbial population in refrigerated raw milk. Different regions and seasons have different levels and types of psychrotrophic bacteria, and these bacteria are highly correlated with the environment, the quality of drinking water and cleaning water, animal feed, and the cleanliness of milking equipment and milking procedures [5, 6]. Therefore, the dairy industry faces challenges resulting from psychrotrophic bacteria that cause poor quality of dairy products. As a result, raw milk bacterial quality is evaluated by measuring levels of psychrotrophic bacteria.

Although psychrotrophic bacteria themselves are not resistant to heat treatments such as pasteurization and ultra-high temperature (UHT) processing, some can produce heat-resistant proteases and lipases that cause spoilage of milk and dairy products upon refrigeration. For example, proteolysis in milk can cause several technological problems such as increased viscosity, age gelation, and bitter flavor in milk and dairy products, as well as a reduction in cheese yield [7–12]. Further concern is the ability of psychrotrophic bacteria to form biofilms on milk tanks and pipelines during storage and transportation. The dairy industry faces several issues related to biofilms, including reduced shelf-life, altered sensorial properties, pathogenicity, and metal corrosion [13–15]. Due to inherent physiological differences and protective shielding provided by extracellular polymeric substances, enzyme production and heat stability are increased in biofilms [16–18].

In light of the major economic impact of proteolytic psychrotrophic bacteria on the dairy industry, these bacteria have been, and continue to be, extensively studied with the main purpose of establishing effective control measures and appropriate regulations to ensure that milk and dairy products are of a high standard. To date, the majority of studies have been performed to identify Gram-negative, proteolytic, psychrotrophic bacteria in raw milk, and to evaluate their capacity to produce proteolytic enzymes and biofilms [3, 19–25]. Although, various genera of Gram-positive bacteria, e.g., *Bacillus*, *Lactococcus*, *Staphylococcus*, *Microbacterium* and *Enterococcus* with the ability to produce biofilm and/or proteolytic enzymes have been identified in raw milk from different geographical regions [20, 26–28], still little is known about the biofilm formation and proteolytic activity of Gram-positive psychrotrophic bacteria, which constitute approximately 10% of the microbial population in cold raw milk that has been stored.

Therefore, in the present study, we first attempted to isolate the Gram-positive, proteolytic, psychrotrophic (G+PP) bacteria from cold raw milk. Next, we identified them based on 16S rRNA gene sequencing, and quantified their proteolytic activity. Finally, we characterized the biofilm-forming capacity of the isolates by determining the biofilm mass using a crystal violet staining assay and quantifying the number of biofilm cells on stainless steel surfaces.

## Materials and methods

### Isolation and identification of Gram-positive, proteolytic, psychrotrophic (G+PP) bacteria from cold raw milk

During winter, a total of 100 samples of cold raw milk were collected in Khuzestan province, in south-western Iran, from different dairy farms and vendors. In this region, the weather is mild in autumn and winter and hot in spring and summer. Therefore, in the months with

moderate temperature, the total bacterial count of raw milk is less. Nevertheless, after cooling milk and during refrigerated storage, the population of psychrotrophic bacteria increases, as expected. Aseptically collected samples were shipped to the lab on ice and analyzed the same day they were taken in sterile bottles over a 6-month period. Each sample was diluted decimally with sterile saline solution, and plated on plate count agar (PCA) supplemented with skim milk powder at a final concentration of 1% (w/v). The plates were incubated at 7°C for 10 days. The presence of a clear zone around the colonies that arises as a result of hydrolysis of the caseins was indicative of proteolysis. Proteolytic colonies (3–5 per plate) with distinguishable morphologies were randomly selected and purified on PCA+1% skim milk. G+PP bacteria were first identified based on Gram-staining and a KOH test, and then by 16S rRNA gene sequencing. The 16S rRNA gene was amplified using universal primers described [29]. The resulting PCR products were extracted from agarose gel using Gel DNA Recovery Kit-EX6151 (SinaClon BioSciences, Iran), and sequenced with a 16S forward primer using a 310 automatic DNA Sequencer (Applied Biosystems, Foster City, CA, USA). A BLAST search was conducted against the 16S rRNA GeneBank database (NCBI) using the partial sequences of the 16S rRNA gene.

## Quantification of the proteolytic activity of the isolates in milk

The identified isolates were inoculated into 10 mL of fresh UHT milk at a final concentration of approximately $10^4$ cfu/mL and incubated at 7°C for 72 h, in order to produce proteolytic enzymes. To determine the proteolytic activity, after incubation the milk samples were centrifuged for 15 min at 5600 g at 4°C. In order to prevent further bacterial growth, the pellet containing the bacterial cells was removed, and sodium azide (0.02%) was added to the supernatant. Then, 1 mL of the supernatant was added to 9 mL of fresh UHT milk in triplicate. One tube was frozen at -20°C (blank) and the others were incubated at 7°C and 25°C for 2 weeks [10, 30, 31].

Hydrolysis of proteins was measured by the O-phthaldialdehyde/N-acetyl-L-cysteine (OPA/NAC) fluorometric assay to determine free amino groups of milk, as explained previously [24, 32]. The fraction of free amino groups (F/F0) was the relative fluorescence unit (RFU) of milk sample after treatment divided by their initial RFU (blank). Accordingly, the isolates were divided into four categories, including very strong (RFU > 2.5), strong (2.5 ≥ RFU > 2), moderate (2 ≥ RFU > 1.5) and weak (1.5 ≥ RFU > 1). The experiments were performed in triplicate.

## Microtitre plate biofilm formation assay

The biofilm-forming ability of G+PP bacteria isolated from cold raw milk was assessed in 96-well polystyrene microtitre plates using crystal violet staining assay, as explained previously [23]. For the comparative analysis of the biofilm-forming ability of the isolates, the method described by Stepanović et al. [33] was used. The cut-off OD ($OD_c$) was defined as three standard deviations above the mean OD of the negative control. All strains were classified into the following categories: Non-producers (N, OD ≤ $OD_c$), low-ability producers (L, $OD_c$ < OD ≤ 2×$OD_c$), moderate-ability producers (M, 2×$OD_c$ < OD ≤ 4×$OD_c$), and high-ability producers (H, 4×$OD_c$ < OD).

## Quantification of biofilm formation on stainless steel surfaces using UHT milk

To evaluate the ability of the identified isolates to produce biofilms on stainless steel surfaces, overnight cultures of each strain were diluted to a final concentration of $10^4$ cfu/mL in UHT

milk. Commercial UHT milk from the same batch was used for all experiments throughout the study. The inoculated UHT milk ($10^4$ cfu/mL, 2 mL) was added into 12-well plates containing $1 \times 1$ cm stainless steel coupons (AISI 304, 2B, Norsk Stål AS, Norway), and incubated at 7˚C without shaking. After 72 h of incubation, the wells were drained completely and the plates were gently washed three times by adding sterile d$H_2O$ (2 mL) to the coupon wells, followed by swirling of the plates and pipetting to remove non-attached bacteria. Attached cells were scraped into 1 mL of physiological saline solution using a cell scraper, and resuspended by vigorous pipetting for 15 s. Colonies were counted on trypticase soy agar (TSA) plates after 36 h of incubation at 25˚C. The experiments were replicated three times on different days [25].

### Statistical analysis

Three separate assays were performed independently, each with three replicates. All data were independent, normally distributed and have the same variances. Results were analyzed using One-way ANOVA, SPSS 20 (SPSS Inc., Chicago, IL). All significance levels are expressed at a 95% confidence level ($p \leq 0.05$).

### Results and discussion

From 100 cold raw milk samples, 343 proteolytic psychrotrophic colonies were isolated and characterized. Gram-staining and the KOH test revealed that the majority of the isolates (n = 307, 89.5%) were Gram-negative bacteria, while only 10.5% (n = 36) were Gram-positive. These Gram-positive isolates, which were originated from 36 different milk samples, were identified based on 16S rRNA gene sequencing. More than 97% similarity level was considered for bacterial species identification. In total, the G+PP bacteria isolated from cold raw milk belonged to 12 genera and 22 species (Table 1). However, 50% of these bacteria belonged to three genera, namely *Staphylococcus* (19.4%), *Bacillus* (16.7%), and *Enterococcus* (13.9%). Vithanage et al. [34] investigated the biodiversity of culturable psychrotrophic microbiota in raw milk, and reported that *Pseudomonas* (19.9%), *Bacillus* (13.3%), *Microbacterium* (5.3%), *Lactococcus* (8.6%), *Acinetobacter* (4.9%) and *Hafnia* (2.8%) were the predominant genera, Subsequently, the identified bacteria were inoculated in fresh UHT milk and incubated at 7˚C for 72 h to produce proteolytic enzyme(s), whose proteolytic activity was then quantified at 7˚C and 25˚C (Table 2). In general, different levels of proteolytic activity were detected for the Gram-positive psychrotrophic isolates. Even among the strains belonging to the same species, the proteolytic activity varied markedly. Furthermore, proteolytic activity was significantly higher at 25˚C than at 7˚C for all isolates ($p<0.05$). At 7˚C, 31 out of 36 G+PP bacteria (86.1%) showed weak proteolytic activity after 2 weeks of incubation, and the remaining 5 (13.9%) showed moderate proteolytic activity. At this temperature, the highest proteolytic activity was observed in *Lactococcus lactis* EL06, *Enterococcus faecalis* EL05, and *Weizmannia coagulans* EL35 (Table 2). However, at 25˚C, only one (2.8%) G+PP bacteria showed weak proteolytic activity; the remaining 15 (41.7%), 14 (38.9%), and 6 (16.6%), showed moderate, strong, and very strong activity, respectively. *L. lactis* EL06, *Microbacterium lacticum* EL30, and *Staphylococcus epidermidis* EL34 showed the highest proteolytic activity (Table 2). Interestingly, among the *Staphylococcus* genus isolates (n = 7), which was the most prevalent G+PP genus, 4 and 1 isolates showed strong and very strong proteolytic activity, respectively.

The ability to produce biofilm is another concern related to psychrotrophic bacteria in milk. This capability can lead to their persistence in milking equipment, tankers, gaskets and pipes in the dairy industry [13–15]. Specifically, the production of enzymes by bacterial cells surrounded by extracellular polymeric substances has previously been demonstrated [35]. These enzymes are either present in the biofilm matrix or dissolved into the surrounding

**Table 1. Putative species identification of the isolates based on 16S rRNA sequence similarities between the isolates and the closest relatives in BLAST.**

| Isolates | Putative species identification | Accession number |
|---|---|---|
| EL01 | Pediococcus pentosaceus | OR268643 |
| EL02 | Lactococcus lactis | OR268644 |
| EL03 | Enterococcus faecium | OR268645 |
| EL04 | Pediococcus pentosaceus | OR268646 |
| EL05 | Enterococcus faecalis | OR268647 |
| EL06 | Lactococcus lactis | OR268648 |
| EL07 | Staphylococcus aureus | OR268649 |
| EL08 | Enterococcus faecium | OR268650 |
| EL09 | Pediococcus acidilactici | OR268651 |
| EL10 | Levilactobacillus brevis | OR268652 |
| EL11 | Staphylococcus epidermidis | OR268653 |
| EL12 | Lactococcus garvieae | OR268654 |
| EL13 | Bacillus licheniformis | OR268655 |
| EL14 | Streptococcus pyogenes | OR268656 |
| EL15 | Lactiplantibacillus plantarum | OR268657 |
| EL16 | Paenibacillus polymyxa | OR268658 |
| EL17 | Bacillus cereus | OR268659 |
| EL18 | Staphylococcus aureus | OR268660 |
| EL19 | Staphylococcus hyicus | OR268661 |
| EL20 | Staphylococcus haemolyticus | OR268662 |
| EL21 | Bacillus mycoides | OR268663 |
| EL22 | Bacillus licheniformis | OR268664 |
| EL23 | Enterococcus faecium | OR268665 |
| EL24 | Weizmannia coagulans | OR268666 |
| EL25 | Microbacterium lacticum | OR268667 |
| EL26 | Bacillus subtilis | OR268668 |
| EL27 | Bacillus thuringiensis | OR268669 |
| EL28 | Paenibacillus polymyxa | OR268670 |
| EL29 | Staphylococcus hyicus | OR268671 |
| EL30 | Microbacterium lacticum | OR268672 |
| EL31 | Pediococcus acidilactici | OR268673 |
| EL32 | Leuconostoc mesenteroides | OR268674 |
| EL33 | Enterococcus faecalis | OR268675 |
| EL34 | Staphylococcus epidermidis | OR268676 |
| EL35 | Weizmannia coagulans | OR268677 |
| EL36 | Lactococcus garvieae | OR268678 |

medium. Researchers examined the diversity of microbiota attached to stainless steel surfaces in a milk processing plant's pre- and post-pasteurization pipeline [20]. They isolated 70 Gram-positive isolates, and identified them as *Enterococcus faecalis* (33 isolates), *Bacillus cereus* (26 isolates), *Staphylococcus hominis* (8 isolates), *Staphylococcus saprophyticus* (2 isolates), and *Staphylococcus epidermidis-Staphylococcus aureus* (1 isolate) species. Among these, 7 strains of *E. faecalis* and 3 strains of *S. hominis* were able to produce biofilm after 24 h of incubation at 37°C. Furthermore, Gram-positive proteolytic bacteria of the genera *Bacillus*, *Staphylococcus*, and *Streptococcus* produced biofilm on stainless steel chips at 25°C [36].

**Table 2. Proteolytic activity of the Gram-positive, proteolytic, psychrotrophic bacteria isolated from raw milk.**

| Identified isolates | RFU (F/F0) [1] | |
| --- | --- | --- |
| | 7°C | 25°C |
| *Pediococcus pentosaceus* EL01 | 1.34 ± 0.12 [ab] | 2.12 ± 0.31 [ab] |
| *Lactococcus lactis* EL02 | 1.28 ± 0.11 [a] | 2.37 ± 0.24 [ab] |
| *Enterococcus faecium* EL03 | 1.21 ± 0.07 [a] | 1.98 ± 0.18 [ab] |
| *Pediococcus pentosaceus* EL04 | 1.35 ± 0.04 [ab] | 1.88 ± 0.35 [ab] |
| *Enterococcus faecalis* EL05 | 1.61 ± 0.17 [b] | 2.56 ± 0.13 [b] |
| *Lactococcus lactis* EL06 | 1.73 ± 0.15 [b] | 2.87 ± 0.17 [b] |
| *Staphylococcus aureus* EL07 | 1.08 ± 0.13 [a] | 1.93 ± 0.21 [ab] |
| *Enterococcus faecium* EL08 | 1.16 ± 0.21 [a] | 1.54 ± 0.26 [a] |
| *Pediococcus acidilactici* EL09 | 1.43 ± 0.09 [ab] | 2.33 ± 0.08 [ab] |
| *Levilactobacillus brevis* EL10 | 1.26 ± 0.12 [a] | 2.18 ± 0.18 [ab] |
| *Staphylococcus epidermidis* EL11 | 1.39 ± 0.17 [ab] | 2.41 ± 0.26 [b] |
| *Lactococcus garvieae* EL12 | 1.48 ± 0.15 [b] | 2.67 ± 0.27 [b] |
| *Bacillus licheniformis* EL13 | 1.31 ± 0.04 [ab] | 2.57 ± 0.31 [b] |
| *Streptococcus pyogenes* EL14 | 1.22 ± 0.07 [a] | 1.67 ± 0.14 [a] |
| *Lactiplantibacillus plantarum* EL15 | 1.19 ± 0.16 [a] | 1.59 ± 0.09 [a] |
| *Paenibacillus polymyxa* EL16 | 1.36 ± 0.18 [ab] | 2.16 ± 0.14 [ab] |
| *Bacillus cereus* EL17 | 1.41 ± 0.21 [ab] | 1.93 ± 0.23 [ab] |
| *Staphylococcus aureus* EL18 | 1.16 ± 0.23 [a] | 2.28 ± 0.14 [ab] |
| *Staphylococcus hyicus* EL19 | 1.43 ± 0.01 [ab] | 2.32 ± 0.28 [ab] |
| *Staphylococcus haemolyticus* EL20 | 1.52 ± 0.13 [b] | 2.26 ± 0.38 [ab] |
| *Bacillus mycoides* EL21 | 1.23 ± 0.11 [a] | 1.67 ± 0.35 [a] |
| *Bacillus licheniformis* EL22 | 1.39 ± 0.18 [ab] | 2.04 ± 0.27 [ab] |
| *Enterococcus faecium* EL23 | 1.19 ± 0.08 [a] | 1.79 ± 0.12 [a] |
| *Weizmannia coagulans* EL24 | 1.28 ± 0.17 [a] | 1.98 ± 0.17 [ab] |
| *Microbacterium lacticum* EL25 | 1.52 ± 0.17 [b] | 2.49 ± 0.31 [b] |
| *Bacillus subtilis* EL26 | 1.14 ± 0.03 [a] | 1.85 ± 0.27 [a] |
| *Bacillus thuringiensis* EL27 | 1.15 ± 0.06 [a] | 1.48 ± 0.14 [a] |
| *Paenibacillus polymyxa* EL28 | 1.24 ± 0.25 [a] | 2.37 ± 0.26 [ab] |
| *Staphylococcus hyicus* EL29 | 1.38 ± 0.21 [ab] | 1.93 ± 0.32 [ab] |
| *Microbacterium lacticum* EL30 | 1.41 ± 0.16 [ab] | 2.75 ± 0.17 [b] |
| *Pediococcus acidilactici* EL31 | 1.29 ± 0.05 [a] | 1.92 ± 0.16 [ab] |
| *Leuconostoc mesenteroides* EL32 | 1.12 ± 0.10 [a] | 1.74 ± 0.18 [a] |
| *Enterococcus faecalis* EL33 | 1.31 ± 0.14 [ab] | 1.89 ± 0.19 [a] |
| *Staphylococcus epidermidis* EL34 | 1.49 ± 0.26 [ab] | 2.69 ± 0.28 [b] |
| *Weizmannia coagulans* EL35 | 1.53 ± 0.06 [b] | 2.44 ± 0.19 [b] |
| *Lactococcus garvieae* EL36 | 1.28 ± 0.10 [a] | 2.22 ± 0.25 [ab] |

[1] Mean±SD values with different small letters in each incubation temperature represent significant difference at $p<0.05$.

However, the biofilm-forming capacity of large collections of Gram-positive psychrotrophic bacterial species has not been investigated, especially at low temperatures. Therefore, in the present study, the biofilm-forming ability of G+PP bacteria was evaluated at 7°C, using a crystal violet staining assay in polystyrene microtitre plates. The cells attached to stainless steel coupons was then quantified.

Results of the crystal violet staining assay are presented in Table 3. Overall, there was marked variation in the biofilm-forming ability of the G+PP bacteria, and also between the

**Table 3. Classification of Gram-positive, proteolytic, psychrotrophic isolates (n = 36) according to their biofilm-forming capacity at 7°C using a crystal violet staining assay.**

| Classification of the isolates | Biofilm formation | | |
|---|---|---|---|
| | 24h | 48h | 72h |
| Non-producers (N) | 19 | 9 | 4 |
| Low-ability producers (L) | 15 | 18 | 10 |
| Moderate-ability producers (M) | 2 | 8 | 19 |
| High-ability producers (H) | 0 | 1 | 3 |
| **Identified isolates** | | | |
| *Pediococcus pentosaceus* EL01 | N | L | L |
| *Lactococcus lactis* EL02 | N | L | M |
| *Enterococcus faecium* EL03 | L | M | H |
| *Pediococcus pentosaceus* EL04 | N | N | M |
| *Enterococcus faecalis* EL05 | N | L | M |
| *Lactococcus lactis* EL06 | N | L | L |
| *Staphylococcus aureus* EL07 | L | L | M |
| *Enterococcus faecium* EL08 | L | L | M |
| *Pediococcus acidilactici* EL09 | M | M | M |
| *Levilactobacillus brevis* EL10 | N | N | N |
| *Staphylococcus epidermidis* EL11 | L | M | H |
| *Lactococcus garvieae* EL12 | N | L | L |
| *Bacillus licheniformis* EL13 | L | M | M |
| *Streptococcus pyogenes* EL14 | N | N | L |
| *Lactiplantibacillus plantarum* EL15 | N | L | L |
| *Paenibacillus polymyxa* EL16 | N | N | N |
| *Bacillus cereus* EL17 | L | M | M |
| *Staphylococcus aureus* EL18 | N | L | M |
| *Staphylococcus hyicus* EL19 | L | L | M |
| *Staphylococcus haemolyticus* EL20 | N | N | L |
| *Bacillus mycoides* EL21 | N | N | N |
| *Bacillus licheniformis* EL22 | M | H | H |
| *Enterococcus faecium* EL23 | L | L | L |
| *Weizmannia coagulans* EL24 | L | M | M |
| *Microbacterium lacticum* EL25 | N | N | N |
| *Bacillus subtilis* EL26 | L | L | M |
| *Bacillus thuringiensis* EL27 | N | L | M |
| *Paenibacillus polymyxa* EL28 | L | M | M |
| *Staphylococcus hyicus* EL29 | N | N | L |
| *Microbacterium lacticum* EL30 | L | L | L |
| *Pediococcus acidilactici* EL31 | N | N | M |
| *Leuconostoc mesenteroides* EL32 | N | L | M |
| *Enterococcus faecalis* EL33 | N | L | M |
| *Staphylococcus epidermidis* EL34 | L | M | M |
| *Weizmannia coagulans* EL35 | L | L | M |
| *Lactococcus garvieae* EL36 | L | L | L |

strains belonging to the same species. After 24 h of incubation, 19 strains (52.8%) did not produce biofilms at 7°C, while 15 (41.7%) and 2 (5.5%) strains produced low and medium amounts of biofilm, respectively. Extension of the incubation time to 48 h decreased the

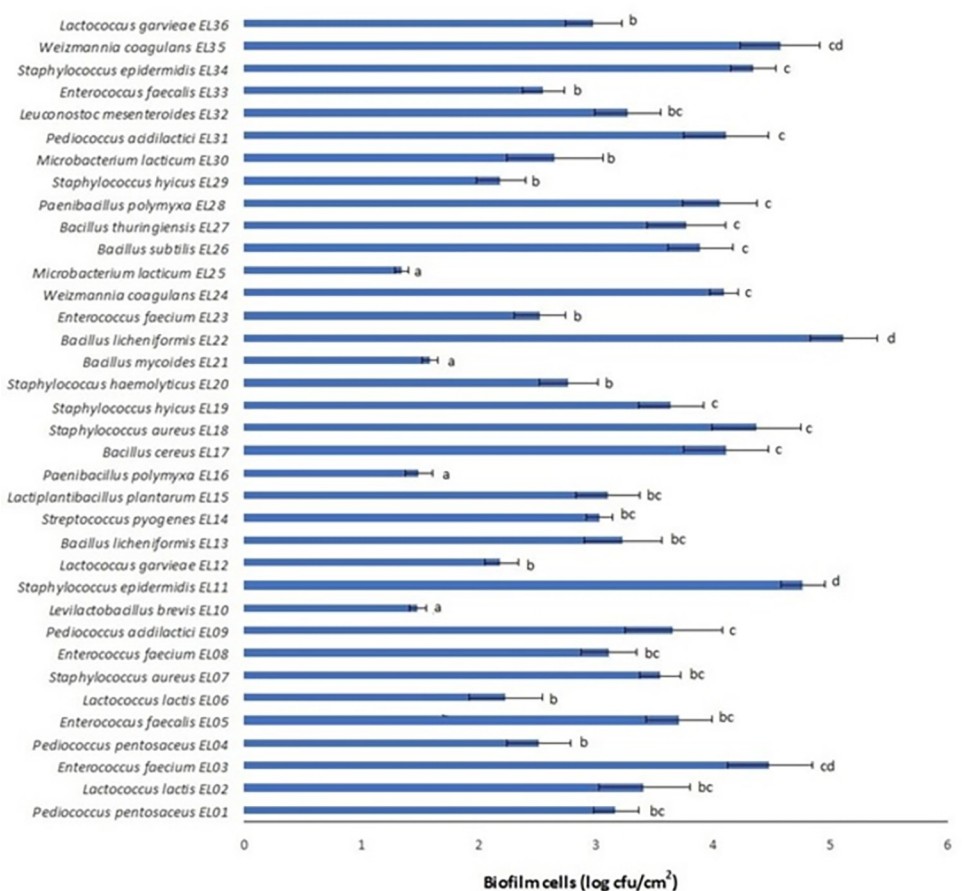

**Fig 1. Biofilm cell counts of the Gram-positive, proteolytic, psychrotrophic bacteria on a stainless steel surface after 72 h incubation at 7°C in UHT milk.**

number of non-producers to 9 (25.0%); in contrast, the numbers of low, moderate and high-ability producers were increased to 18 (50.0%), 8 (22.2%) and 1 (2.8%) strains, respectively. At the end of the 72-h incubation period, 4 strains (11.1%), namely *Levilactobacillus brevis* EL10, *Paenibacillus polymyxa* EL16, *Bacillus mycoides* EL21, and *M. lacticum* EL25 did not produce a biofilm at 7°C, while 10 (27.8%) and 19 (52.8%) strains produced low and moderate amounts of biofilm, respectively. At this time, 3 strains (8.3%), namely *Enterococcus faecium* EL03, *S. epidermidis* EL11, and *B. licheniformis* EL22 produced high amounts of biofilm on polystyrene microplates.

Since stainless steel is mainly used for processing surfaces, equipment, and tools in dairy manufacturing plants, the ability of the G+PP bacteria isolated from cold raw milk to produce biofilms on stainless steel surfaces was also evaluated. As shown in Fig 1, all of the 36 G+PP bacteria were able to produce a biofilm on the surface of stainless steel coupons in UHT milk. The number of attached cells ranged from 1.34 to 5.11 log cfu/cm$^2$. Among the G+PP bacteria, the highest number of biofilm cells on the stainless steel surface belonged to *B. licheniformis* EL22 (5.11±0.29 log cfu/cm$^2$), *S. epidermidis* EL11 (4.76±0.19 log cfu/cm$^2$), and *W. coagulans* EL35 (4.57±0.34 log cfu/cm$^2$). It has previously been reported that *B. licheniformis* adhere to stainless steel surfaces (6.12 log cfu/cm$^2$) [37].

## Conclusion

Although G+PP bacteria are less prevalent in cold raw milk than their Gram-negative counterparts, their ability to produce biofilms and proteolytic enzymes can lead to their persistence in dairy processing equipment, and ultimately spoilage of milk. This study is one of the few to provide information on the proteolytic activity and biofilm-forming ability of Gram-positive psychrotrophic bacteria in raw milk. We found that proteolytic activity and biofilm-forming capacity of G+PP bacteria isolated from cold raw milk are strain-dependent, and different levels of these attributes can be detected in different strains. In addition, these results are remarkable because they were performed in a real-world environment, i.e., milk, and at a low temperature, hence they accurately reflect the conditions in the dairy industry.

## Supporting information

**S1 Table. Some microbial characteristics of the milk samples used in this study.**
(DOCX)

## Author Contributions

**Conceptualization:** Mehdi Zarei, Siavash Maktabi.

**Data curation:** Siavash Maktabi.

**Formal analysis:** Mehdi Zarei, Siavash Maktabi, Amin Yousefvand.

**Funding acquisition:** Mehdi Zarei, Siavash Maktabi, Per Erik Joakim Saris.

**Investigation:** Sahar Elmi Anvari, Siavash Maktabi, Amin Yousefvand.

**Methodology:** Sahar Elmi Anvari, Siavash Maktabi, Amin Yousefvand.

**Supervision:** Mehdi Zarei.

**Writing – original draft:** Mehdi Zarei, Siavash Maktabi, Amin Yousefvand.

**Writing – review & editing:** Mehdi Zarei, Siavash Maktabi, Per Erik Joakim Saris.

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
