## [Decision Letter · Decision Letter 0]

5 Jul 2023

PONE-D-23-16130Identification, proteolytic activity quantification and biofilm-forming characterization of Gram-positive, proteolytic, psychrotrophic bacteria isolated from cold raw milkPLOS ONE

Dear Dr. Zarei,

Thank you for submitting your manuscript to PLOS ONE. After careful consideration, we feel that it has merit but does not fully meet PLOS ONE’s publication criteria as it currently stands. Therefore, we invite you to submit a revised version of the manuscript that addresses the points raised during the review process.

 Please consider the comments of both reviewers

We look forward to receiving your revised manuscript.

Kind regards,

Guadalupe Virginia Nevárez-Moorillón, Ph.D.

Academic Editor

PLOS ONE

https://linkinghub.elsevier.com/retrieve/pii/S0958694620301576

https://dokumen.pub/microbiology-in-dairy-processing-challenges-and-opportunities-9781119114802-1119114802-9781119114970-1119114977-9781119114987-1119114985.html

In your revision ensure you cite all your sources (including your own works), and quote or rephrase any duplicated text outside the methods section. Further consideration is dependent on these concerns being addressed.

“A research grant (SCU.VF99.245) provided by Shahid Chamran University of Ahvaz supported this study. In addition, The Finnish Food Research Foundation and The Finnish Society of Sciences and Letters are acknowledged for awarding grants to Amin Yousefvand on the dates 16.03.2021 and 28.03.2022. The open access funded by Helsinki University Library.”

Reviewers' comments:

Reviewer's Responses to Questions

**Comments to the Author**

1. Is the manuscript technically sound, and do the data support the conclusions?

Reviewer #1: Yes

Reviewer #2: Yes

2. Has the statistical analysis been performed appropriately and rigorously? 

Reviewer #1: I Don't Know

Reviewer #2: Yes

3. Have the authors made all data underlying the findings in their manuscript fully available?

Reviewer #1: No

Reviewer #2: Yes

4. Is the manuscript presented in an intelligible fashion and written in standard English?

Reviewer #1: Yes

Reviewer #2: No

5. Review Comments to the Author

Reviewer #1: The topic of the manuscript (Gram-positive psychrotrophic bacteria in raw milk, their proteolytic activity and biofilm-forming abilities) has not been well recognized until now, therefore the results presented in the manuscript would be of interest to the readers. The manuscript presents in a clear manner the subject, the methodological part is described clearly and mostly detailed enough. The biggest disadvantage of the manuscript is the lack of any characteristics of the milk samples taken for analysis. I understand that the main goal of the work was focused on the isolation and characteristics of the G+PP bacteria, however, at least basic information on the raw material should be added e.g. total bacterial number, numbers of mesophilic and psychroptophic bacteria, proportions of Gram-positive to Gram-negative bacteria in the population of the psychrotrophic ones. This information (particularly the latter one) would enable the Authors to elaborate on the importance of their work for the dairy industry.

Other missing information is listed below, and it should be included in the manuscript before considering the manuscript for publication.

16S rRNA gene sequences should be deposited in a public repository, e.g. in GenBank and accession numbers should be placed in the manuscript

Line 83-84 Please give information on the season when the samples were taken and if it influences the bacterial composition of raw milk in Iran. You can include this information here or in the discussion section.

Line 124 Please give the inoculation level

Line 133-136 Please give information on statistical tests applied for data analysis, include also the requirements that data had to meet to be processed with the tests.

Line156 Please give information on whether the 36 G+PP isolates were from different sampling locations/milk samples. Please discuss the prevalence of G+PP bacteria among psychrotrophic bacteria and their (G+PP bacteria) possible impact on the milk production process.

Reviewer #2: Reviewer Comments

Manuscript PONE-D-23-16130

In this study, the authors aimed to isolate

and characterize Gram-positive, proteolytic and psychrotrophic bacteria from raw milk. The enzymatic activities and formation of biofilms are very exploited abilities in Gram-negative contaminants of milk, which can be considered a differential of the present study. The contextualization of the problem in the Introduction does not bring anything new and is weakly supported by outdated references. Only 17% (4) of the references are from the last five years, and the most recent are from 2020. It would be interesting if the authors addressed in the Introduction the reason that led them to analyze Gram-positive bacteria, which according to the authors, constitute approximately 10% of the microbial population in cold raw milk that has been stored. The experimental strategy is simple, adopts basic and conventional techniques for isolating and forming biofilms, and uses molecular techniques (16S rRNA) for isolate identification.

General Comments

- An update of bibliographic references is strongly needed. The subject is widely studied, and there are many recent publications.

- Standardization is required. For example, there are two forms of volume units: ml and mL.

Specific Comments

Line 62: sensorial properties instead of organoleptic properties

Lines 71-73: Why are these few studies on Gram-positive psychrotrophic bacteria not mentioned here? This is the subject of this study and was not explored in the Introduction. What are the genera of proteolytic

psychrotrophic bacteria found in milk? Do they form biofilms?

Information about it is not considered in the context of the addressed problem.

Line 91: Was the sequence not analyzed from 16S rDNA? How was the DNA extracted? Inform about the methods, reagents, or kits used and the referred commercial brands.

Line 101: The centrifugation detail should be informed in centrifuge force (g), not rpm.

Lines 138-150: All this information belongs to the Introduction section. There are no results or discussion here.

Line 140: Please check the accuracy of this information in the cited references.

Line 190: A thorough literature review should be done before making this statement

Line 198: In many situations, the biofilm production by certain bacteria is only verified when the substrate is periodically changed.

Line 210: Although there is no consensus on the number of cells present on the surface to be considered a biofilm, this value of 1.34 log cfu/cm2 is too low and is not considered a biofilm.

Line 237: A careful standardization of the reference section is necessary.

6. PLOS authors have the option to publish the peer review history of their article (what does this mean?). If published, this will include your full peer review and any attached files.

Reviewer #1: No

Reviewer #2: No

---

## [Author Response · Author response to Decision Letter 0]

2 Aug 2023

Responses to the Respected Editor and Reviewers' Comments:

Editor's comment:

Comment 1: Please ensure that your manuscript meets PLOS ONE's style requirements, including those for file naming. The PLOS ONE style templates can be found at

Response 1: Thank you so much. It was checked again and carefully. 

Comment 2: We noticed you have some minor occurrence of overlapping text with the following previous publication(s), which needs to be addressed:

https://linkinghub.elsevier.com/retrieve/pii/S0958694620301576

https://dokumen.pub/microbiology-in-dairy-processing-challenges-and-opportunities-9781119114802-1119114802-9781119114970-1119114977-9781119114987-1119114985.html

In your revision ensure you cite all your sources (including your own works), and quote or rephrase any duplicated text outside the methods section. Further consideration is dependent on these concerns being addressed.

Response 1: Thank you so much. You have mentioned an article and a book. The mentioned article, which is one of my own articles, is in the reference list of this manuscript. Before submitting this manuscript, I did my best to include that text in a rephrased form. If there are still some sentences that need to be rephrased, I'd appreciate it if you could let me know those sentences.

However, in the case of the mentioned book, that book was not used in writing this manuscript. Maybe there was a sentence from somewhere else or in my mind as the author, the main source of which is this book, but I did not know about it. If there are some sentences that need to be rephrased, I'd appreciate it if you could let me know those sentences.

Comment 3: We suggest you thoroughly copyedit your manuscript for language usage, spelling, and grammar. If you do not know anyone who can help you do this, you may wish to consider employing a professional scientific editing service.

Response 3: Thank you so much. Before submitting this manuscript to PLOS One, we submitted that to Language revision services at the University of Helsinki. (https://www.helsinki.fi/en/language-centre/cooperation-and-language-services/language-revision-services-at-the-university-of-helsinki).

They edited the manuscript for English writing. I will submit the edited manuscript as a supporting information file. 

Comment 4: Thank you for stating the following in the Acknowledgments Section of your manuscript:

“A research grant (SCU.VF99.245) provided by Shahid Chamran University of Ahvaz supported this study. In addition, The Finnish Food Research Foundation and The Finnish Society of Sciences and Letters are acknowledged for awarding grants to Amin Yousefvand on the dates 16.03.2021 and 28.03.2022. The open access funded by Helsinki University Library.”

Response 4: Funding information was removed from the acknowledgment and added to the cover letter.

Comment 5: In your Data Availability statement, you have not specified where the minimal data set underlying the results described in your manuscript can be found. PLOS defines a study's minimal data set as the underlying data used to reach the conclusions drawn in the manuscript and any additional data required to replicate the reported study findings in their entirety. All PLOS journals require that the minimal data set be made fully available. For more information about our data policy, please see http://journals.plos.org/plosone/s/data-availability.

Response 5: Supplementary information files were prepared and submitted.

Comment 6: Please include a separate caption for each figure in your manuscript.

Response 6: A separate caption for the figure was included in the manuscript.

Reviewers' comments:

Reviewer #1

Comment 1: The topic of the manuscript (Gram-positive psychrotrophic bacteria in raw milk, their proteolytic activity and biofilm-forming abilities) has not been well recognized until now, therefore the results presented in the manuscript would be of interest to the readers. The manuscript presents in a clear manner the subject, the methodological part is described clearly and mostly detailed enough. 

Response 1: Thank you so much.

Comment 2: The biggest disadvantage of the manuscript is the lack of any characteristics of the milk samples taken for analysis. I understand that the main goal of the work was focused on the isolation and characteristics of the G+PP bacteria, however, at least basic information on the raw material should be added e.g. total bacterial number, numbers of mesophilic and psychroptophic bacteria, proportions of Gram-positive to Gram-negative bacteria in the population of the psychrotrophic ones. This information (particularly the latter one) would enable the Authors to elaborate on the importance of their work for the dairy industry.

Response 2: Thanks for your good suggestions. Unfortunately, we do not have information about the total number and the number of mesophilic bacteria in the milk samples because the collection of this information was not in line with the goals of our research. As you know, the focus of this study was on psychrotrophic bacteria, so the incubations were done only for this category of bacteria. We have the information on the proportions of Gram-positive to Gram-negative bacteria in the population of the psychrotrophic bacteria. However, considering that this article specifically deals with the subject of psychrotrophic, proteolytic bacteria and the ratio of psychrotrophic proteolytic gram-positive bacteria to psychrotrophic proteolytic gram-negative bacteria has presented in the results section, in order to avoid confusion of the readers and interference of topics, the desired information of the respected reviewer is presented as supplementary information. I hope it will be accepted by the respected reviewer.

Comment 3: 16S rRNA gene sequences should be deposited in a public repository, e.g. in GenBank and accession numbers should be placed in the manuscript

Response 3: Thanks for your good suggestions. Based on this comment, table 1 was revised and the desired information was added

Comment 4: Line 83-84 Please give information on the season when the samples were taken and if it influences the bacterial composition of raw milk in Iran. You can include this information here or in the discussion section.

Response 4: Thank you for your good suggestions. All the samples were collected during winter. This information was added to the text.

Comment 5: Line 124 Please give the inoculation level

Response 5: Thank you so much. The inoculation level was added to the text.

Comment 6: Line 133-136 Please give information on statistical tests applied for data analysis, include also the requirements that data had to meet to be processed with the tests.

Response 6: Thank you so much. This information was added to the text.

Comment 7: Line156 Please give information on whether the 36 G+PP isolates were from different sampling locations/milk samples. Please discuss the prevalence of G+PP bacteria among psychrotrophic bacteria and their (G+PP bacteria) possible impact on the milk production process.

Response 7: Thank you so much. This information was added to the text.

Reviewer #2

Comment 1: In this study, the authors aimed to isolate and characterize Gram-positive, proteolytic and psychrotrophic bacteria from raw milk. The enzymatic activities and formation of biofilms are very exploited abilities in Gram-negative contaminants of milk, which can be considered a differential of the present study. The contextualization of the problem in the Introduction does not bring anything new and is weakly supported by outdated references. Only 17% (4) of the references are from the last five years, and the most recent are from 2020. It would be interesting if the authors addressed in the Introduction the reason that led them to analyze Gram-positive bacteria, which according to the authors, constitute approximately 10% of the microbial population in cold raw milk that has been stored. 

Response 1: Thanks for your good suggestions. Introduction of the manuscript was revised based on this comment. 

Comment 2: An update of bibliographic references is strongly needed. The subject is widely studied, and there are many recent publications.

Response 2: The entire manuscript was revised accordingly and recent publications were added to the text. 

Comment 3: Standardization is required. For example, there are two forms of volume units: ml and mL.

Response 3: It was corrected in the text.

Comment 4: Line 62: sensorial properties instead of organoleptic properties

Response 4: It was corrected in the text.

Comment 5: Lines 71-73: Why are these few studies on Gram-positive psychrotrophic bacteria not mentioned here? This is the subject of this study and was not explored in the Introduction. What are the genera of proteolytic psychrotrophic bacteria found in milk? Do they form biofilms? Information about it is not considered in the context of the addressed problem.

Response 5: Thanks for your good suggestions. Introduction was revised accordingly and new information was added.

Comment 6: Line 91: Was the sequence not analyzed from 16S rDNA? How was the DNA extracted? Inform about the methods, reagents, or kits used and the referred commercial brands.

Response 6: Thank you so much. The resulting PCR products were extracted from agarose gel using Gel DNA Recovery Kit-EX6151 (SinaClon BioSciences, Iran), and sequenced with a 16S forward primer using a 310 automatic DNA Sequencer (Applied Biosystems, Foster City, CA, USA). This information was added to the text.

Comment 7: Line 101: The centrifugation detail should be informed in centrifuge force (g), not rpm.

Response 7: Thank you so much. It was corrected in the text.

Comment 8: Lines 138-150: All this information belongs to the Introduction section. There are no results or discussion here.

Response 8: Thanks for your good suggestion. The text was revised accordingly.

Comment 9: Line 140: Please check the accuracy of this information in the cited references.

Response 9: Thank you so much. This information was deleted in the revised manuscript.

Comment 10: Line 190: A thorough literature review should be done before making this statement

Response 10: Your point is absolutely right. We again searched the available sources. indeed, very few studies have been done on the biofilm formation of Gram-positive psychrotrophic bacteria. 

Comment 11: Line 198: In many situations, the biofilm production by certain bacteria is only verified when the substrate is periodically changed.

Response 11: My apologies, but I don't understand what the respected reviewer means and its relevance to this research. If it is stated more clearly, it will be considered in the revised manuscript.

Comment 12: Line 210: Although there is no consensus on the number of cells present on the surface to be considered a biofilm, this value of 1.34 log cfu/cm2 is too low and is not considered a biofilm.

Response 12: Thank you so much. I understand your point, but I draw your attention to the fact that all plates were washed three times before quantifying the biofilm. Therefore, this low number of bacteria are definitely attached to the surface and so, we should consider them as biofilms.

Comment 13: Line 237: A careful standardization of the reference section is necessary.

Response 13: Thank you so much. References were checked again and carefully.

---

## [Decision Letter · Decision Letter 1]

15 Aug 2023

PONE-D-23-16130R1Identification, proteolytic activity quantification and biofilm-forming characterization of Gram-positive, proteolytic, psychrotrophic bacteria isolated from cold raw milkPLOS ONE

Dear Dr. Zarei,

Thank you for submitting your manuscript to PLOS ONE. After careful consideration, we feel that it has merit but does not fully meet PLOS ONE’s publication criteria as it currently stands. Therefore, we invite you to submit a revised version of the manuscript that addresses the points raised during the review process.

There are some suggestions to be incorporated into the manuscript. Please revise the document.  Please submit your revised manuscript by Sep 29 2023 11:59PM. If you will need more time than this to complete your revisions, please reply to this message or contact the journal office at plosone@plos.org. Please include the following items when submitting your revised manuscript:A rebuttal letter that responds to each point raised by the academic editor and reviewer(s). You should upload this letter as a separate file labeled 'Response to Reviewers'.A marked-up copy of your manuscript that highlights changes made to the original version. You should upload this as a separate file labeled 'Revised Manuscript with Track Changes'.An unmarked version of your revised paper without tracked changes. You should upload this as a separate file labeled 'Manuscript'.If applicable, we recommend that you deposit your laboratory protocols in protocols.io to enhance the reproducibility of your results. Protocols.io assigns your protocol its own identifier (DOI) so that it can be cited independently in the future. For instructions see: https://journals.plos.org/plosone/s/submission-guidelines#loc-laboratory-protocols. Additionally, PLOS ONE offers an option for publishing peer-reviewed Lab Protocol articles, which describe protocols hosted on protocols.io. Read more information on sharing protocols at https://plos.org/protocols?utm_medium=editorial-email&utm_source=authorletters&utm_campaign=protocols.

We look forward to receiving your revised manuscript.

Kind regards,

Guadalupe Virginia Nevárez-Moorillón, Ph.D.

Academic Editor

PLOS ONE

Journal Requirements:

Reviewers' comments:

Reviewer's Responses to Questions

**Comments to the Author**

1. If the authors have adequately addressed your comments raised in a previous round of review and you feel that this manuscript is now acceptable for publication, you may indicate that here to bypass the “Comments to the Author” section, enter your conflict of interest statement in the “Confidential to Editor” section, and submit your "Accept" recommendation.

Reviewer #1: (No Response)

2. Is the manuscript technically sound, and do the data support the conclusions?

Reviewer #1: Yes

3. Has the statistical analysis been performed appropriately and rigorously? 

Reviewer #1: Yes

4. Have the authors made all data underlying the findings in their manuscript fully available?

Reviewer #1: Yes

5. Is the manuscript presented in an intelligible fashion and written in standard English?

Reviewer #1: (No Response)

6. Review Comments to the Author

Reviewer #1: Ref. to Comment 4: “Please give information on the season when the samples were taken and if it influences the bacterial composition of raw milk in Iran. You can include this information here or in the discussion section.” Please comment on how the season of milk collection influences the microbiological composition of milk in Iran. If there is no data on this, please also include this information.

Line 56 “poor quality” of what?

Line 153 Please add information on the % similarity to the species reference sequence obtained for sequences obtained in this study. You may give the threshold of the %similarity applied for the species identification. You may give the information here or in Table 1.

Line 158 “predominant genera. Subsequently”

Supplementary Information – please give the units of the data. Is it colony forming units per milliliter?

7. PLOS authors have the option to publish the peer review history of their article (what does this mean?). If published, this will include your full peer review and any attached files.

Reviewer #1: No

---

## [Author Response · Author response to Decision Letter 1]

16 Aug 2023

Responses to the Respected Reviewers' Comments:

Comment 1: Ref. to Comment 4: “Please give information on the season when the samples were taken and if it influences the bacterial composition of raw milk in Iran. You can include this information here or in the discussion section.” Please comment on how the season of milk collection influences the microbiological composition of milk in Iran. If there is no data on this, please also include this information.

Response 1: Thank you so much. In this region, the weather is mild in autumn and winter and hot in spring and summer. Therefore, in the months with moderate temperature, the total bacterial count of raw milk is less. nevertheless, after cooling milk and during refrigerated storage, the population of psychrotrophic bacteria increases, as expected. This information was added to the text.

Comment 2: Line 56 “poor quality” of what?

Response 1: Thank you so much. It was corrected in the text: poor quality of dairy products.

Comment 3: Line 153 Please add information on the % similarity to the species reference sequence obtained for sequences obtained in this study. You may give the threshold of the %similarity applied for the species identification. You may give the information here or in Table 1.

Response 3: Thank you so much. More than 97% similarity level was considered for bacterial species identification. This information was added to the text.

Comment 4: Line 158 “predominant genera. Subsequently”

Response 4: Thank you so much. It was corrected. 

Comment 5: Supplementary Information – please give the units of the data. Is it colony forming units per milliliter?

Response 5: Thank you so much. Yes, it is CFU/ml. It was corrected in the Supplementary Information.

---

## [Editor Report · Decision Letter 2]

21 Aug 2023

Identification, proteolytic activity quantification and biofilm-forming characterization of Gram-positive, proteolytic, psychrotrophic bacteria isolated from cold raw milk

PONE-D-23-16130R2

Dear Dr. Zarei,

We’re pleased to inform you that your manuscript has been judged scientifically suitable for publication and will be formally accepted for publication once it meets all outstanding technical requirements.

Kind regards,

Guadalupe Virginia Nevárez-Moorillón, Ph.D.

Academic Editor

PLOS ONE
---

## [Editor Report · Acceptance letter]

6 Sep 2023

PONE-D-23-16130R2 

Identification, proteolytic activity quantification and biofilm-forming characterization of Gram-positive, proteolytic, psychrotrophic bacteria isolated from cold raw milk 

Dear Dr. Zarei:

I'm pleased to inform you that your manuscript has been deemed suitable for publication in PLOS ONE. Congratulations! Your manuscript is now with our production department. 

Kind regards, 

on behalf of

Dr. Guadalupe Virginia Nevárez-Moorillón 

Academic Editor

PLOS ONE